# Spatiotemporal Mislocalization of Nuclear Membrane-Associated Proteins in γ-Irradiation-Induced Senescent Cells

**DOI:** 10.3390/cells9040999

**Published:** 2020-04-17

**Authors:** Svobodová Kovaříková Alena, Bártová Eva, Kovařík Aleš, Lukášová Emilie

**Affiliations:** 1Laboratory of Molecular Cytology and Cytometry, Institute of Biophysics, Czech Academy of Sciences, Královopolská 135, 61265 Brno, Czech Republic; alenakovarikova@ibp.cz (S.K.A.); bartova@ibp.cz (B.E.); 2Laboratory of Molecular Epigenetics, Institute of Biophysics, Czech Academy of Sciences, Královopolská 135, 61265 Brno, Czech Republic; kovarik@ibp.cz; 3Laboratory of Cell Biology and Radiobiology and Laboratory of Molecular Epigenetics, Institute of Biophysics, Czech Academy of Sciences, Královopolská 135, 61265 Brno, Czech Republic

**Keywords:** lamin B receptor, LINC complex proteins, nesprin-1, SUN1/2, γ-irradiation, heterochromatin-nuclear membrane disconnection

## Abstract

Cellular senescence, induced by genotoxic or replication stress, is accompanied by defects in nuclear morphology and nuclear membrane-heterochromatin disruption. In this work, we analyzed cytological and molecular changes in the linker of nucleoskeleton and cytoskeleton (LINC) complex proteins in senescence triggered by γ-irradiation. We used human mammary carcinoma and osteosarcoma cell lines, both original and shRNA knockdown clones targeting lamin B receptor (LBR) and leading to LBR and lamin B (LB1) reduction. The expression status and integrity of LINC complex proteins (nesprin-1, SUN1, SUN2), lamin A/C, and emerin were analyzed by immunodetection using confocal microscopy and Western blot. The results show frequent mislocalization of these proteins from the nuclear membrane to cytoplasm and micronuclei and, in some cases, their fragmentation and amplification. The timing of these changes clearly preceded the onset of senescence. The LBR deficiency triggered neither senescence nor changes in the LINC protein distribution before irradiation. However, the cytological changes following irradiation were more pronounced in shRNA knockdown cells compared to original cell lines. We conclude that mislocalization of LINC complex proteins is a significant characteristic of cellular senescence phenotypes and may influence complex events at the nuclear membrane, including trafficking and heterochromatin attachment.

## 1. Introduction

The nuclear envelope in an interphase cell is anchored internally to the scaffold of the nucleoskeleton and chromatin and externally to the cytoskeleton. The nuclear pores and proteins of the inner nuclear membrane (INM) interact with each other and with the nucleoskeleton, while protein bridges span the perinuclear space linking proteins of the INM with proteins in the outer nuclear membrane (ONM), which in turn bridge to the cytoskeleton. The complex of proteins called the LINC (linker of nucleoskeleton and cytoskeleton) complex connects the perinuclear space with the nucleoskeleton and cytoskeleton [1,2,3,4]. This LINC complex plays a role in maintaining nuclear morphology, nuclear positioning, gene expression, and cell signaling [4,5,6,7]. Its major protein components include a family of multiisomeric scaffolding nesprins, SUN1/SUN2, emerin, and lamin A/C [1,8,9,10]. Lamin A/C can interact with DNA and core histones, while emerin interacts with lamin A/C and DNA-associated proteins, such as BAF [11,12]. Extensive alternative transcription and splicing of genes encoding nuclear envelope proteins containing spectrin repeat (*SYNE1* and *SYNE2)* genes generate multiple spectrin-repeat isoforms that vary greatly in size and exhibit multiple subcellular localization, especially the nesprins-1 and -2 isoforms [2]. The typical structure of giant nesprins-1 and -2 consists of three major domains: a C-terminal KASH domain that is targeted to the nuclear envelope (NE), an N-terminal paired Calponin Homology (CH) domain which binds to the actin cytoskeleton, and a central rod domain containing multiple spectrin repeats (SRs), which links the CH and KASH domains of the molecule [3]. The giant isoforms localize in the ONM and interact, by means of the KASH domain, with SUN1 and SUN2 at the perinuclear space, in this way forming the LINC complex that connects the nucleus to actin cytoskeleton. Nesprin-3 interacts via plectin with intermediate filaments, small nesprins isoforms, like nesprin-1α, lacking the CH domain at the N terminal, and nesprin-4 also localize in the ONM, forming LINC with microtubules via interactions with dynein and microtubule motor protein kinesin-1 in the cytoplasm. Small nesprin isoforms can also localize in the INM [1,3]. Nesprins-1/2 are ubiquitously expressed and are highly abundant in skeletal and cardiac muscles, in particular, smaller isoforms nesprins-1α_2_ and nesprins-2α_1_ [1,13]. KASH-less nesprin variants have been identified in multiple cytoplasmic and nuclear compartments [3]. Mutation of the LINC complex proteins may lead to numerous pathophysiological conditions, namely in cardiac and skeletal muscles. These histological types are known to harbor a rich system of LINC complex proteins [14]. In Emery–Dreifuss muscular dystrophy (EDMD) patients, these mutations lead to defects in nuclear morphology and nucleoskeletal uncoupling, as studied in fibroblasts [15,16,17,18,19]. Thus, LINC complex mutations are likely to have an effect on NE integrity, resulting in the uncoupling of the nucleoskeleton and cytoskeleton [20,21,22]. We recently found that DNA damage induced by γ-irradiation or replication stress (RS) in cancer cells leads to downregulation of the lamin B receptor (LBR) and lamin B1 (LB1) associated with changes in nuclear morphology [23,24]. LBR is an integral protein of the inner nuclear membrane (INM) which preferentially binds to LB1 at the N terminal [25]. Its main function is to tether heterochromatin to the nuclear membrane in embryonic and non-differentiated cells [26]. Interestingly, the changes that we observed in nuclear morphology were similar to those described in fibroblasts and myoblasts from Emery–Dreifuss muscular dystrophy (EDMD) and cardiomyopathy (CMP) [15]. The reduction of LBR and LB1 induced by γ-irradiation was accompanied by the uncoupling of heterochromatic regions from the nuclear membrane and their distension in nucleoplasm in epithelial and fiborsarcoma cells [23].

It is widely accepted that DNA damage induced by different stresses results in irreversible alterations of chromatin structure and function, leading to the cessation of cell proliferation and cellular senescence [27,28,29]. Relatively little is known about the distribution of LINC proteins in senescent cells and the effects of irradiation on the integrity of the nuclear membrane. Therefore, we decided to investigate the behavior of LINC complex proteins (nesprin-1, SUN1/2), emerin, and LA/C in actively proliferating and γ-irradiated cells doomed to senescence. Additionally, we looked at the influence of LBR/LB1 reduction on the potential mislocalization of LINC proteins in the nuclear membrane. For this study, we used two cancer cells lines of different histological origin, both wild-type and shRNA knockout targeting LBR. The integrity and quantity of proteins were analyzed by Western blot. 

## 2. Material and Methods

### 2.1. Cell Culture

Human cell lines of mammary carcinoma MCF7 (ATCC collection, HTB-22), osteosarcoma U2OS (ATTC HTB-96), brain glioblastoma U-87 (ATCC HTB-14), colon colorectal adenocancer HT29 (ATCC 38), and lung carcinoma A549 (ATTC CRM-CCL-185) were used. The cells were grown in Dulbecco’s modified Eagle’s medium with 10% fetal bovine serum (Gibco, Thermo Fisher Scientific, Waltham, MA, USA), 100 U/mL penicillin, and 0.1 mg/mL streptomycin (Sigma-Aldrich, St. Louis, MO, USA). The MCF7 and U2OS cells with constitutively reduced levels of lamin B receptor (called MCF7-LBR(-) and U2OS-LBR(-) thereafter) were prepared by transformation of cells with plasmid-based small hairpin RNA (shRNA) (Sigma-Aldrich) constructs targeting LBR [23]. These knockdown clones were selected on antibiotics and further grown in normal media. Silencing of LBR was stable over at least 4 years of cultivation and did not change during repeated freezing/thawing cycles. They were grown in Dulbecco’s modified Eagle’s medium, the same as original MCF7 and U2OS cell lines, however, their proliferation was slightly lower compared to the maternal cells. All cells were grown at 37 °C and 5% carbon dioxide (CO_2_).

### 2.2. Cell Irradiation

Irradiation was performed using a ^60^Co γ-ray source at Chizostat (Chirana, CR). Cells were seeded at a density of 2 × 10^5^/mL and irradiated 24 h later in culture medium at 37 °C under normal atmospheric conditions with 8 Gy (D = 1 Gy/min). Cells were irradiated in 25 mm^2^ culture vessels to assess the levels of certain proteins and on slides in four-well dishes (Nunc, #167063, Thermo Scientific, Rochester, NY, USA) for the immunodetection of DNA double strand breaks (DSBs), heterochromatin markers, lamins, LBR, nesprin-1, SUN1/2, emerin, and SA-β-gal (senescence-associated β-galactosidase) activity. After irradiation, the cells were incubated at 37 °C and 5% CO_2_ until further treatment, with replacement of the growth medium every other day. Cells were grown attached to the bottom of microscopic slides immersed in growth medium, and replaced every second day, for approximately two weeks. They became senescent from day 7 PI (post-irradiation), with many having their nucleus broken up into a number of micronuclei, blisters, and other nuclear morphology. However, these cells remained attached to the microscopic slide, which distinguished the senescent from the apoptotic cells that peeled off immediately from the vial bottom and were removed when changing media. 

### 2.3. Antibodies and Immunofluorescence

Microscopic slides containing cells were withdrawn at different time intervals from the culture medium, washed 2× in phosphate-buffered saline (PBS; 140 mM NaCl, 2.7 mM KCl, 1.5 mM KH_2_PO_4_, and 6.5 mM Na_2_HPO_4_; pH 7.2) at 37 °C, and fixed in 4% paraformaldehyde in PBS for 10 min at 22 °C. The cells were then rinsed briefly in PBS, washed three times for 5 min in PBS, permeabilized in 0.2% Triton X-100/PBS for 15 min at room temperature, and washed twice in PBS for 5 min. Before incubation with primary antibodies (overnight at 4 °C), the cells were blocked with 5% inactivated fetal calf serum + 2% bovine serum albumin/PBS for 30 min at room temperature. Antibodies from two different hosts were used on each slide to detect two different antigens in the same nuclei. Anti-H2AX phosphorylated at serine 139 #05-636, anti-53BP1 #4937, and anti-β-actin #4970 antibodies were obtained from Cell Signaling (Dellaertweg 9b, 2316 WZ Leiden, The Netherlands). Anti-lamin B1 #ab8982 and anti-lamin B receptor #ab32535 antibodies were obtained from Abcam (Cambridge Biomedical Campus, Cambridge, CB2 OAX, UK). Anti-lamin A/C #SAB4200236 was from Sigma-Aldrich (Companies House, London 2204655, UK), anti-emerin H-12 #sc-25284 was from Santa Cruz Biotechnology (Inc. 101 Cooper St, Santa Cruz, CA, USA), Nesprin-1 monoclonal antibody (Mannes1A(7A12) #MA5-18077 was from Thermo Fisher (Thermo-Fisher-Scientific-Life Technologies, Brno, Czech Republic), SUN1 antibody #NBP1-87396 was from Novus Biological (19 Barton Lane, Abingdon Science Park, Abingdon, OX14 3NB, UK), and SUN2 antibody #HPA001209 was from Atlas Antibodies (Voltavagen 13A, SE-168 69 Bromma, Sweden). The secondary antibodies, affinity purified-FITC conjugated donkey anti-mouse and affinity purified Cy3-conjugated donkey anti-rabbit, were obtained from Jackson Immuno-Research Laboratories (West Grove, PA, USA). After the slides were preincubated with 5% donkey serum/PBS for 30 min at room temperature, a mixture of both antibodies was applied to each slide and incubated for 1 h in the dark at room temperature. This incubation was followed by washing three times for 5 min each in PBS. Cells were counterstained with 1 mM TOPRO-3 (Molecular Probes, Eugene, OR, USA) in 2× saline sodium citrate (SSC) prepared fresh from stock solution. After the cells were briefly washed in 2× SSC, Vectashield medium (Vector Laboratories, Burlingame, CA, USA) was used for the final mounting of the samples.

### 2.4. Confocal Fluorescence Microscopy of Spatially Fixed Cells

The immunofluorescence of the detected proteins was analyzed using images obtained with a high-resolution Leica DM RXA confocal microscope (Leica, Wetzlar, Germany) equipped with an oil immersion Plan Fluotar objective (100×/NA1.3) and a CSU 10a Nipkow disk (Yokogawa, Japan) for confocal imaging. A CoolSnap HQ CCD-camera (Photometrix, Tuscon, AZ, USA) and an Ar/Kr laser (Innova 70C Spectrum, Coherent, Santa Clara, CA, USA) were used for image acquisition. Automated exposure, image quality control, image analysis, and other procedures were performed using Acquiarium software [30]. The exposure time and dynamic range of the camera in the red, green, and blue channels were adjusted to the same values for all slides to obtain quantitatively comparable images. Forty serial optical sections were captured at 0.2 mm intervals (along the *z*-axis). In total, 100–300 cells were recorded for each set of conditions, and the experiments were repeated three times. The results are reported as standard error of mean. A *t*-test was used for the statistical comparison of specified effects obtained in cell nuclei after γ-irradiation and before irradiation.

### 2.5. Senescence-Associated β-Galactosidase Assay

Detection of senescence-associated β-galactosidase (SA-β-gal) activity [31] was performed in control non-irradiated cells and 24 h, 72 h, and 7 days after cell exposure to γ-irradiation using a Senescence Detection Kit #K320-250 from Bio-Vision Incorporated (Milpitas, CA, USA), according to the manufacturer’s instructions. Before using the kit, the cells were fixed in 4% paraformaldehyde for 10 min at room temperature and washed three times in PBS. Images were captured using an Olympus BX51 microscope equipped with an Olympus DP72 camera and Quick Photo Micro 2.3 software at 200× magnification.

### 2.6. SDS-PAGE and Western Blot

Cells were washed in PBS, scraped in the presence of Complete Mini EDTA-free protease inhibitors (Roche Diagnostics, #04693159001) and a cocktail of phosphatase inhibitors PhosSTOP (Roche Diagnostics, #04906845001) and centrifuged. Cells were washed in PBS twice and lysed in 200 μL of 1% SDS. Lysates were frozen and afterwards sonicated for 8 s at maximum power. Protein concentrations were modified by a μQuantTM microplate spectrophotometer (Biotek, Winooski, VT, USA). Then, 5 μL of a 1:1 β-mercaptoethanol/bromphenol blue mixture was added to lysates which were then boiled for 5 minutes. Proteins were separated according to their size by SDS-PAGE and afterwards, transferred to a polyvinylidene fluoride (PVDF) membrane (#10600021, GE Healthcare Life Sciences, Little Chalfont, UK). Settings for electrophoresis and proteins’ transfer was performed according to Franek et al. [32]. Membranes for detection of proteins were incubated over night at 4 °C in blocking solution (2% milk) with the following primary antibodies: anti-emerin H-12 (#sc-25284, Santa Cruz Biotechnology, dilution 1:3000), anti-nesprin-1 (MA5-18077, ThermoFisher Scientific, dilution 1:500), anti-lamin B1 (#ab8982, Abcam, dilution 1:500), anti-lamin B receptor (#ab32535, Abcam, dilution 1:1000), anti-lamin A/C (#SAB4200236, Sigma-Aldrich dilution 1:1500), anti-SUN1 (#NBP1-87396 Novus Biological, dilution 1:2000), anti-SUN2 (#HPA001209 Atlas Antibodies, dilution 1:1000), anti-H3 (#ab7091, Abcam, dilution 1:100,000), anti-H3K9me3 (#ab8898, Abcam, dilution 1:2000), anti-α-tubulin (#ab80779, Abcam, dilution 1:1500), anti-actin (c-11) (#sc-1615, Santa Cruz Biotechnology dilution 1:500), anti-GAPDH (#sc-365062 Santa Cruz Biotechnology, dilution 1:1000), and anti-UBF (#sc-9131, Santa Cruz Biotechnology, dilution 1:1000 in 2% gelatin). As secondary antibodies, the following were used: anti-rabbit IgG (#A-4914, Merck, Germany; dilution 1:2000), anti-mouse IgG (#A-9044, Merck, Germany; dilution 1:2000), and anti-mouse IgG1 (#sc-2060, Santa Cruz Biotechnology, Dallas, TX, USA; dilution 1:1000). Proteins were visualized by the ECL Western blotting detection reagent (#RPN2106, GE Healthcare Life Sciences, Little Chalfont, UK) on a LAS-3000 system (Fujifilm, Tokyo, Japan)

## 3. Results

### 3.1. Changes in Nuclear Morphology of MCF7 and U2OS Cancer Cell Lines after Irradiation 

We used a high dose of γ-rays to reach high genotoxic stress, leading to non-reparable DNA damage and transition of a large number of irradiated cells to senescence. Changes in nuclear morphologies in relation to LB1 and LBR protein levels were followed at different times after irradiation with a dose of 8 Gy (Figure 1A,B). The levels of LBR and LB1 had already decreased significantly 24 h after the irradiation (24 h PI) (Figure 1C,D, Appendix A). The lower response in U2OS compared to MCF7 cells can be explained by a relatively high level of LBR in U2OS cells [23]. At 24 h PI, cells did not express SA-β-galactosidase while they abundantly expressed it at day 7 to day 12 PI (Figure 2). Nevertheless, the important changes in nuclear morphology and in LINC complex proteins’ localization already appear together with the reduction of LBR and LB1 at 24 h PI. These changes present the rapid response of cells to DNA damage (Figure 1A,B). The frequency of morphological defects observed at later PI time intervals are given in Appendix A. Results show that the nuclear morphology was changed in about 61% of MCF7 cells at 24 h PI. Multiple changes in nuclear morphology occurred simultaneously in the same cell. About 33% of cells formed micronuclei, 33% exhibited a convoluted appearance, 18% contained blisters, 4.5% presented tweens connected with anaphase bridges, and 2.4% of cells were fragmented (Figure 1A, Figure 3, Appendix A). Only minor morphological changes represented by small micronuclei in about 6% of cells were observed in control cancer MCF7 cells before irradiation (Figure 1A, Appendix A). The number of nuclei containing one or more morphologic changes increased with time after irradiation and the number of cells with normal morphology decreased (Appendix A). About 84% of cells exhibited micronuclei, 42% exhibited convoluted nuclei, 26.5% exhibited blisters, 16% nuclei were fragmented, 10% had anaphase bridges, 91% cells lost LBR, and 80% lost LB1 at 72 h PI. Later, at 7 days PI, the majority of cells were senescent and displayed the most intense changes in nuclear morphology. At this time, almost 99% of cells lost LBR and 90% lost LB1, 82% of cells formed micronuclei, 57% formed blisters, 44% were convoluted, 19% were fragmented, and 7% had anaphase bridges (Appendix A).

Control U2OS cells also had small defects in nuclear morphology before irradiation, consisting of the presence of small micronuclei in about 7% of cells and honeycombing in 1.2% of cells, indicating the disruption of the nuclear membrane (Figure 1B, Figure 3, Appendix A). Similar to MCF7, U2OS cells already manifested changes in nuclear morphology in 60% of cells at 24 h PI (Figure 1B, Appendix A). Most cells (33%) presented micronuclei, 28% of nuclei were giant, 22% of nuclei formed blisters, and 17% formed convoluted nuclei. Changes in nuclear morphology increased with time in all cells. About 66% of cells presented micronuclei, 63% were convoluted, about 56% formed blisters, 15% cells were fragmented, 11% exhibited honeycombing, and the same amount had giant nuclei at 72 h PI (Appendix A). Contrary to MCF7, U2OS cells did not form anaphase bridges, while MCF7 did not present cells with giant nuclei after irradiation. There were no cells without morphological changes and such changes in nuclear morphology were very complex after a longer time period (Appendix A). In U2OS cells, there was 5% honeycombing of the nuclei with interrupted NM and protrusion of chromatin into the cytoplasm and 39% of cells with giant nuclei 7 days PI. The most frequent morphological changes (both 71%) of U2OS cells were the formation of micronuclei and nuclei with a convoluted appearance, whilst fragmented nuclei formed 47% at this time (Appendix A, Figure 3).

Changes in nuclear morphology were also observed in MCF7, and U2OS cells deficient in LBR (LBR(-)). Both cell types showed slower proliferation but did not evolve senescent phenotypes. In these cells, the LBR and LB1 immunostaining signals were only 20% and 28% respectively, of those of control cells and decreased further after irradiation with 8 Gy of γ-rays (Figure 1C,D). In general, the LBR(-) cells also exhibited more abnormalities in nuclear morphology after the γ-irradiation than the original cells (Figure 1A–D, Appendix A).

### 3.2. Defects in Localization of Emerin, LA/C, SUN1, SUN2, and Nesprin-1 Cells Exposed to γ-Irradiation.

We used confocal microscopy in combination with immunohistochemical staining to analyze localization of LINC proteins in actively growing cells and cells after the irradiation. In the non-irradiated cells (both original and LBR(-)), emerin and SUN2 colocalized with LA/C in the inner nuclear membrane (INM). In contrast, the antibody against SUN1 showed a dispersed signal throughout the whole nucleus, especially in heterochromatin. The nesprin-1 signal was present in the nucleus as well as the cytoplasm (Appendix A). 

After the irradiation, we observed mislocalization of emerin, LA/C, SUN1, SUN2, and nesprin-1 in different forms to cytosol and micronuclei (MN). Comparisons of frequencies of different phenotypic changes occurring 24 h after the irradiation are shown in Figure 4A,B. These changes were frequent, especially in LBR(-) cells, showing early response of cells to DNA damage by genotoxic stress. Changes appearing at later intervals (72 h and 7 days post irradiation) are shown in Appendix A. In MCF7 cells, emerin dispersed in the cytosol in almost all cells, often forming clumps. In addition, emerin was frequently (46% nuclei) incorporated into micronuclei (Figure 4A). Emerin, together with LA/C, colocalized in heterochromatic MN or formed clumps in the cytosol in about 30% of nuclei. LA/C manifested minor dots in cytosol and bordered MN in about 9% of cells (Figure 4A, Figure 5A–C, Appendix A, Appendix A).

SUN1 formed clumps in cytosol in about 22% of nuclei and was concentrated to polar caps in about 5% of nuclei. However, SUN2 was most frequently concentrated in MN and formed clumps in cytosol, together in 45% of nuclei. Nesprin-1 formed small clumps in cytosol or polar caps in about 28% of nuclei (Figure 4A, Appendix A). SUN2 together with nesprin-1 filled heterochromatic MN in a small proportion of nuclei.

In U2OS cells, in addition to emerin, changes also occurred in SUN 1, namely the relocation of clumps to cytosol and the accumulation to micronuclei in about 31% of nuclei at 24 h PI (Figure 4B, Appendix A). Emerin was dispersed in the cytosol of almost all cells, similarly to in the MCF7 cell line and in the MCF7 LBR(-) and U2OS LBR(-) cells (Figure 4A,B, Appendix A). This protein formed polar caps or clumps in the cytosol of about 33% of nuclei. Emerin was also frequently dispersed in the cytosol together with LA/C in a high number of nuclei. Nesprin-1 formed small cytosol clumps in about 10% of cells (Figure 4B). Minor clumps of SUN2 were present in about 5% of cells. SUN2 together with nesprin-1 formed polar caps in 4.4% of the U2OS cell line. Numerous changes in the mislocalization and their frequencies were observed at later (72 h and 7 days PI) periods after irradiation, especially in the U2OS cell line (Appendix A). These senescent cells fully expressing SA-β-gal regularly showed disappearance of emerin, SUN1, and SUN2 from the nuclear membrane while they formed clumps in the cytosol or filled MN (Appendix A). 

There was a greater variability in the type and frequency of NM proteins’ mislocalizations in shRNA knockdown cells after the irradiation. This was particularly evident for SUN1 protein (Figure 4A, Appendix A) where numerous (90% MCF7-LBR(-) cells) defects in its distribution in the 24 h interval of the post irradiation period. In contrast, only relatively infrequent (27% of cells) small clumps were observed in the cytosol and in MCF7 cells. These defects included protein amplification in the NE and its liberation to cytosol in the form of different sized clumps in about 32% of MCF7-LBR(-) cells, relocation to MN, with minor clumps at anaphase bridges in about 58% of these cells. The SUN2 protein showed less frequent mislocalization than SUN1 at the cytological level and their frequency was similar between MCF7 and MCF7-LBR(-) cells (Figure 4A, Appendix A). A similar degree of dispersion of emerin in the cytosol was observed in both MCF7 and MCF7-LBR(-) cells (Appendix A).

The differences in frequencies of nuclear membrane protein defects between control and LBR(-) cells were also notable in the U2OS line (Figure 4B, Appendix A). Here, the most profound defect was the relocation of the SUN2 protein forming only individual clumps in the cytosol of 5% of control U2OS cells, while it exhibited various forms of the mislocalizations in nearly 100% of U2OS-LBR(-). Amplification of SUN2 in the NM was apparently the reason for its dispersion to the surroundings of the nucleus in about 50% of cells and relocation of larger fragments to the cytosol (disintegration of this protein) of 50% of LBR(-) cells. These forms, together with the filling of MN of 38% of LBR(-) cells, indicate the presence of more different defects of this protein in the same cell. Large differences were also noted in the location of different forms of SUN1 defects between U2OS and U2OS-LBR(-). Specifically, amplification of this protein and its consecutive disintegration and clumps in cytosol occurred in about 60% of U2OS-LBR(-) nuclei followed by mere amplification in the NE and the filling of MN in 10% of U2OS-LBR(-). On the other hand, clumps in the cytosol were observed only in about 27% of nuclei in irradiated U2OS cells. Similarly, as in MCF7, aberrant localization of emerin was more frequent (75%) in U2OS-LBR(-) than in U2OS (33%) cells. The phenotypic changes included protein clumps in the cytosol, polar caps, fillings in MN, and segmental fragmentation in the NE, sometimes together with LA/C. The more frequent changes in localization of nesprin-1 were also observed in the U2OS-LBR(-) (24%) compared U2OS (10%) cells. Together, these observations show that the reduced level of LBR and LB1 in LBR(-) cells likely decreases the strength of the nuclear membrane and facilitates the release and degradation of residing proteins. 

Of note, the LBR-deficient cells showed not only greater incidence of these defects but also their greater diversity. For example, single irradiated cells exhibiting different defects in the same protein were significantly more frequent in LBR(-) cells compared to original cell lines (*p* ≤ 0.05). Examples of these multiple defects in LINC complex proteins in original cells are shown in Figure 5A–C and in LBR(-) cells in Appendix A. Therefore, it is more precise to analyze all defects of a given protein in order to determine the fate of a given protein in irradiated cells. 

### 3.3. Expression and Integrity of LINC Proteins Analyzed by Western Blot

The level of LINC proteins as well as other proteins known to be involved in cellular responses to irradiation were investigated by Western blots (Figure 6A–C). The proteins were assayed in control cells 24 h, 72 h, and 7 days post irradiation (PI) in MCF7 and U2OS cells. The U2OS cells showed a reduction of SUN2 and actin proteins at 72 h PI and in senescent cells (day 7 PI) but were not observed in MCF7 cells (Figure 6A). On the other hand, the level of nesprin-1 was increased in senescent MCF7 cells (day 7 PI) compared to the control and earlier post irradiation periods. Overall, there were either minor or no actual changes in LINC protein expression in irradiated cells and the differences in individual protein levels seem to be cell-type-specific. However, in both cell lines, there was a consistent decrease in H3K9m3 and UBF (45S rDNA transcription factor) at later time intervals. 

Inspection of signals in whole blots revealed substantial fractionalization of nesprin-1 and SUN2 proteins in both cell lines (Figure 6B,C). Other proteins showed no or marginal fragmentation. Intact nesprin-1 migrated as a band in the 250 kDa region in both cell lines (Figure 6B). These bands were visible in both control and irradiated U2OS and MCF7 cells. Low molecular weight fragments of 125 and 110 kDa began to appear after 72 h PI. The 125 kDa fragment was more pronounced in U2OS than in MCF7 cells. The senescent MCF7 cells showed a relatively strong 250 kDa band corresponding to intact nesprin-1.

The intact SUN2 protein migrated in the 85 kDa region in both cell lines (Figure 6C). In the U2OS cells, a 70 kDa fragment was visualized starting from 72 h, and its presence was accompanied by a concomitant decrease of the intensity of the 80 kDa band. Fragments of SUN2 migrating as 53 and 40 kDa bands were already visible in the control proliferating MCF7 cells. However, their intensity was low while progressively increasing after the irradiation. The 40 kDa fragment was the most dominant band in senescent cells (day 7 PI). The LBR deficiency is documented by near absence of very weak bands on blots in U2OS-LBR(-) and MCF-LBR(-) cells (Appendix A).

## 4. Discussion

While the effects of irradiation on DNA and chromatin topology are relatively well described, little is known about the impact of γ-radiation on nuclear membrane integrity and its protein components. Here, we analyzed changes in LINC complex proteins (nesprin-1, SUN1, and SUN2), emerin, and LA/C under genotoxic stress induced by gamma irradiation. We observed profound changes in their localization, accompanied by changes in nuclear morphology already starting at early intervals (24 h) after the irradiation. The changes occurred in a high proportion of cells of different histological origin. We also showed that the LBR/LB1 reduction induced by cell irradiation appears as a characteristic marker of cells determined to be headed to senescence after exposure to genotoxic stress [23,24].

### 4.1. Early Changes in Localization of LINC Proteins in Irradiated Cells

In irradiated cells, we detected the dispersion of LINC complex proteins, emerin, and LA/C in the cytoplasm, where they often formed clumps. In addition, they were found in micronuclei or aberrantly amplified in the nuclear membrane. All these cytological changes, collectively called mislocalization, can be considered as a specific response to irradiation since little or none of them were found in actively proliferating non-irradiated cells. Defects in nuclear morphology and LINC complex protein distribution were also observed in the human lung carcinoma cell line A549, colon cancer cell line HT29, and human glioblastoma cell line U87 after irradiation (not shown), generally corroborating our findings for solid tumors. Mislocalization of LINC proteins started to be visible as early as 24 h after the irradiation, coinciding with the reduction of LBR and LB1 levels, both preceding the onset of senescence by at least 72 h. Thus, there seems to be a significant lag phase between changes in the nuclear membrane and the expression of SA-β-galactosidase, indicating that the transition to senescence was relatively slow [31]. Though we can only speculate about cellular processes accompanying this period, we observed a progressive fragmentation of nesprin-1 and SUN2 proteins whose intensity was the most pronounced 7 days following irradiation, suggesting preferential degradation of mislocalized proteins during the senescence pathway. The activation of specific proteases targeting mislocalized LINC proteins cannot be excluded. In our experiments, not all membrane proteins were similarly affected by irradiation. For example, emerin seems to be regularly dispersed into the cytoplasm in a majority of cells, irrespective of the cell type. One explanation could be its relatively small size (about 30 kDa) compared to other LINC proteins, facilitating its mobilization in the cell. In contrast, nesprin-1 displayed relatively little changes at the cytological level. This can be attributed to its large size (up to several thousands of Daltons in some isoforms) and the presence of multiple proteins binding to the N- or C-terminal of this protein, including SUN1/SUN2 [2,3,4,33]. Indeed, we frequently observed the colocalization of nesprin-1 with SUN1/2 at aberrant sites in the cytoplasm and micronuclei (Figure 4, Figure 5, Appendix A, Appendix A). SUN1 and nesprin-1 exhibited cell-type-specific responses to genotoxic stress showing distinct cell-type-specific fragmentation on immunoblots. In contrast, reduction of UBF (an RNA Polymerase I transcription factor) was observed in advanced stages of senescence in both cell types, which is consistent with epigenetic reprogramming of nucleoli after the UV light irradiation [34]. Loss of heterochromatic H3K9me3 could be associated with detachment of heterochromatin from NM and its distension in senescent cells’ nucleoplasm [23].

Cellular senescence has been established as a cellular response to a variety of stresses, such as DNA double strand breaks and oncogene activation [27,28,29]. However, a link between DNA damage and defects in localization of LINC proteins remains to be established. Double strand breaks difficult to repair make apparently decision about irreversible exit from the cell cycle during 3–4 h after their induction [33]. Cell cycle exit together with high activity of p53 and p21 after irradiation [23] are probably sufficient to drive cells to senescence. We envisage that these complex events trigger downregulation of LBR/LB1 expression, detachment of heterochromatin from NM, and mobilization of LINC proteins. Previous research indicated involvement of LINC complex proteins in DSB repair [35,36,37,38]. Certainly, we cannot exclude a possibility that at least some types of mislocalization we observed here could be attributed to an active role of LINC proteins in DNA repair processes. 

### 4.2. LBR/LB1 Deficiency May Enhance LINC Proteins Mobilization in Irradiated Cells

In our previous study, we showed that the irradiation of MCF7 and U2OS cells induced LBR/LB1 downregulation accompanied by the detachment of centromeric satellite heterochromatin from the nuclear membrane, its relocation to the nucleoplasm, and distension, which was consistent with LBR/LB1 reduction [23]. Here, we explored the possibility of whether the reduction of these proteins would affect the localization of LINC proteins in the same cell lines. However, except perhaps for the slightly increased fraction of cells with mislocalized emerin in U2OS cells, the LBR/LB1 deficiency did not induce significant changes in localization of LINC proteins. This suggests that normally proliferating cancer cells may not need LBR to maintain the correct position of LINC proteins in the nuclear membrane and that the function of these proteins differ from that of nesprin-1, whose knockdown in mesenchymal stems cells apparently induced severe morphological changes in the nucleus, such as fusion and fragmentation. In another study, knockdown of nesprin-1 resulted in an increased number of DNA breaks [35,39]. Whether the LBR deficiency induces higher frequency of DNA breaks is currently unknown. In any case, the LBR/LB1 reduction by shRNA led to an increased frequency of LINC protein mislocalization after irradiation, perhaps due to a decreased strength of nuclear envelope caused by a lower level of LBR and LB1. 

### 4.3. Possible Clinical Implications of Elevated LINC Protein Mobility in Irradiated Cells 

Several authors [5,9,14,15,16,17,18,19,21,40,41,42,43,44] described defects in nuclear morphology and mislocalization of LINC complex proteins in cells of patients with autosomal dominant EDMD as well as CMP caused by mutations in nesprins-1/2 and other LINC complex proteins. However, they did not connect these defects with senescence. We hypothesize that mutations in LINC complex proteins leading to mislocalization of these proteins accompanied by changes in nuclear morphology may induce cellular senesce in heart and skeletal muscle cells, similarly to non-reparable DNA damage induced by γ-irradiation in cancer cells. Senescence in cancer cells was manifested by similar defects in LINC complex proteins’ localization and nuclear morphology, such as in cells of skeletal and heart cells, where these defects were considered by some authors [5,43,44] as the cause of nucleoskeleton cytoskeleton uncoupling that may also affect dysregulation of gene expression. These opinions are in accordance with our observation of changes in chromatin structure due to disconnection of heterochromatin sequences from nuclear envelope in senescent cells exposed to γ-irradiation [23]. It is thus very probable that mutation in genes coding for LINC complex proteins on one side and DNA damage induced by genotoxic stress on the other side lead to cellular senescence accompanied by the disconnection of chromatin from the NE. While the impact of mutations in nesprin and LINC protein is manifested mainly in myocytes and cardiomyocytes, rich in these proteins, the consequences of genotoxic stress are particularly evident in solid tumors after exposure to different stresses. However, defects in the localization of LINC proteins, emerin, and LA/C, as well as changes in nuclear morphology, occur in both cases.

## Figures and Tables

**Figure 1 cells-09-00999-f001:**
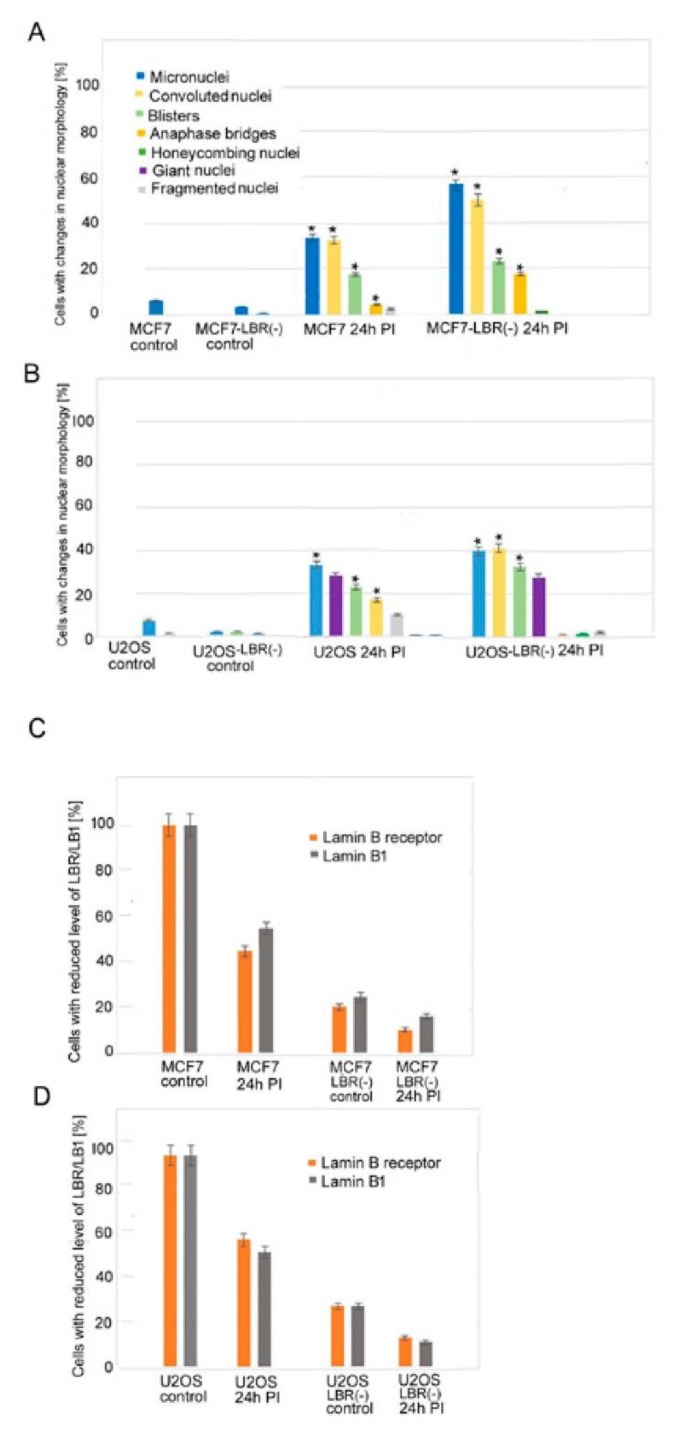
Graphical presentation of changes in nuclear morphology after the irradiation of cells with 8 Gy of γ-rays (**A**,**B**). Levels of LBR and LB1 in original cell lines and LBR(-) cells stably expressing LBR-specific shRNA (LBR-deficient clones) (**C**,**D**). (**A**) Defects in original MCF7 and LBR(-) cells. (**B**) Defects in original U2OS cell line and LBR(-) cells. (**C**) Reduction of LBR and LB1 levels in MCF7 cell line and LBR(-)-cells. (**D**) Reduction of LBR and LB1 levels in U2OS cell line and LBR(-)cells. The data were collected from 3 independent experiments. More than 100 cells were counted in each group. The results represent a percentage of the mean number of cells ± SE. There are significant (student *t-*test, *p* ≤ 0.05, asterisks) differences in the incidence of most defects between LBR(-) and original cell lines calculated for the comparison of MCF7-LBR(-) with the MCF7 cell line in incidence of micronuclei (MN), convoluted cells, cells with blisters, and cells with anaphase bridges. For comparison of U2OS-LBR(-) with the U2OS cell line, significant (*p* ≤ 0.05) difference were found in the incidence of MN, convoluted cells, and number of cells with blisters.

**Figure 2 cells-09-00999-f002:**
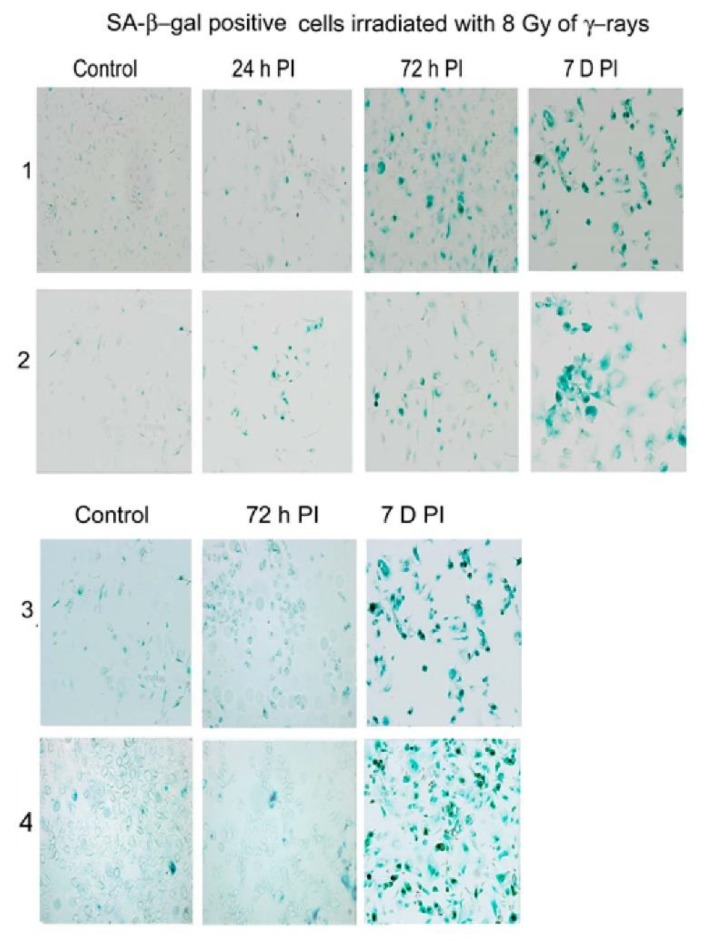
Activity of SA-β-galactosidase in MCF7, U2OS, MCF7-LBR(-) and U2OS-LBR(-) cells after the irradiation with 8 Gy of γ-rays. (1) MCF7, (2) U2OS, (3) MCF7-(LBR-), (4) U2OS-LBR(-). PI—post irradiation.

**Figure 3 cells-09-00999-f003:**
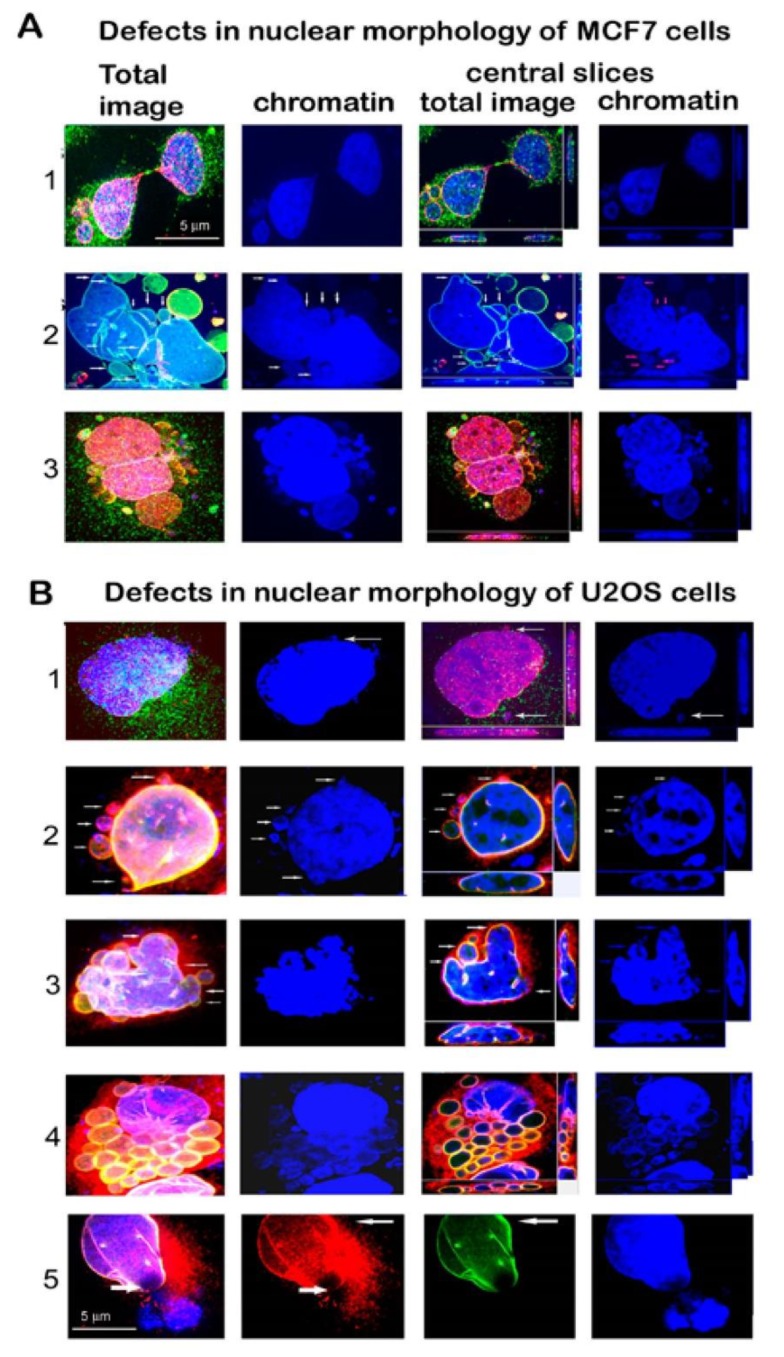
Examples of nuclear morphology defects in (**A**) MCF7 and (**B**) U2OS cell lines 24 h PI with the dose of 8 Gy of γ-rays. (**A**) 1—An anaphase bridge linking the separating daughter nuclei, SUN1 (red), nesprin1 (green), 2—Convoluted nucleus with blisters (arrows), LA/C (green), emerin (red). 3—Fragmented nucleus represented by 3 large and numerous small micronuclei. SUN2 (red), nesprin-1 (green). (**B**) 1— A giant nucleus with small chromatin fragments and micronuclei (MN, arrows), SUN1 (red), nesprin-1 (green). 2—Nucleus with micronuclei, emerin (red), lamin A/C (green), arrows indicate MN containing emerin and heterochromatin (HC), 3—Convoluted nucleus with blisters (arrows), emerin (red), LA/C (green). 4—Fragmented nucleus, emerin (red), LA/C (green). 5—Honeycombed nucleus, emerin (red), LA/C (green), thick arrow shows interrupted NE enabling chromatin to leak out of the nucleus, reinforced layer of LA/C (green) and emerin (red), (thin arrows). Size bar indicates µm.

**Figure 4 cells-09-00999-f004:**
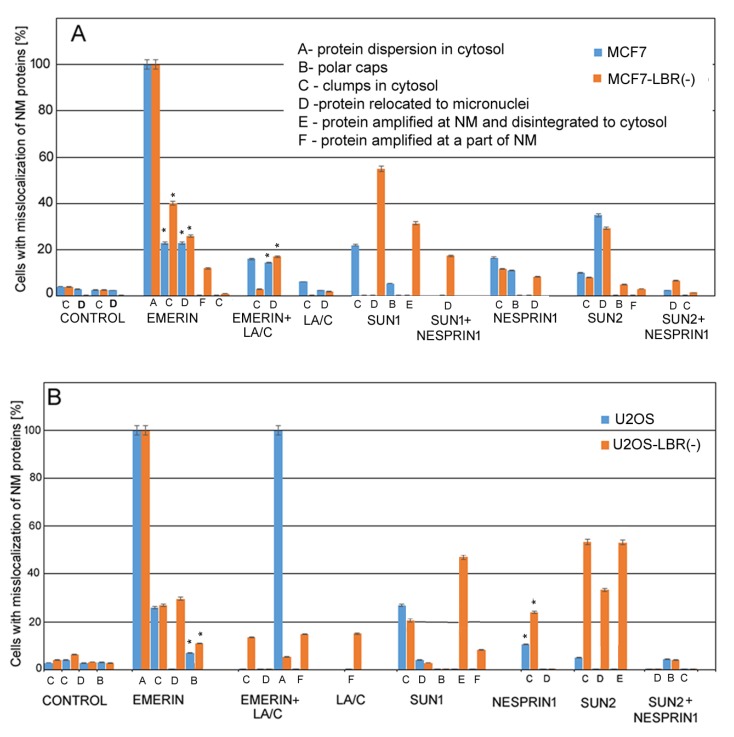
Graphical presentation of mislocalization frequency of LINC complex proteins (nesprin-1, SUN1, SUN2), emerin, and LA/C in irradiated cells. (**A**) MCF7 cell line and derived LBR(-) cells, (**B**) U2OS cell line and derived LBR(-) cells. The data are presented as a percentage from the mean cell number ± SE. The differences between parental cells and derived clones were significant for almost all LINC complex proteins. The data were collected from 3 independent experiments and more than 100 cells were counted in each. In less obvious cases, differences between LBR(-) and original cells were statistically evaluated (*t*-test, *p* ≤ 0.05, asterisks).

**Figure 5 cells-09-00999-f005:**
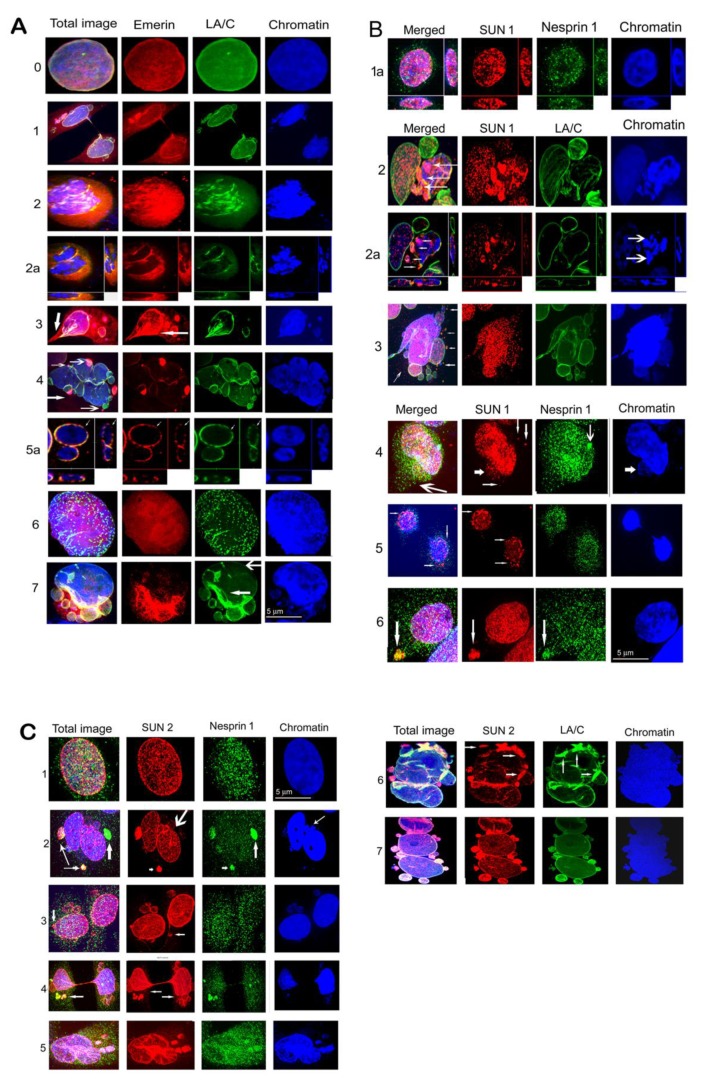
(**A**) Examples of mislocalization of emerin in MCF7 and U2OS cells irradiated with 8 Gy of γ-rays 24 h PI. 0—Control MCF7 cell, 1—MCF forming anaphase bridge and MN, some of MN are filled with emerin, which is also dispersed in the cytosol. 2—U2OS divided into 2 nuclei located in close proximity. A part of emerin is dispersed in the cytosol and another part forms short fragments together with LA/C at the NE. Chromatin is condensed into fragments resembling senescence-associated heterochromatic foci (SAHF) in senescent cells, better distinguishable on 2a presenting a central slice through figure 2 and showing that lamin A/C together with emerin form an interrupted layer in the NE of these two nuclei and both proteins are partially dispersed into the cytosol. 3—MCF7 with dispersed emerin in the cytosol, some MN are also filled with emerin and heterochromatin. In addition, emerin is amplified in a region of the NE (arrow). The nucleus has a section of an anaphase bridge in the lower left corner (thick arrow). 4—Convoluted nucleus of MCF7 with emerin forming polar caps. A layer of emerin is weakened in the NE, LA/C binds nuclei and MN. Its layer is also weakened at some positions in the NE. 5a—central slice through MCF7 nuclei shows interrupted LA/C (green) and emerin (red) in the NE (thin arrows). 6—Giant U2OS nucleus with fragmented LA/C, emerin is dispersed in the nucleus. 7—U2OS-LBR(-) with reinforced LA/C (green) and emerin (red) at a lower part of the nucleus (narrow arrow). Emerin is dispersed in the cytoplasm and in MN in this region. Both proteins are lost at the upper part of the nucleus (wide arrow). Size bar indicates 5 µm. (**B**). Examples of SUN1 and nesprin-1 mislocalization in MCF7 and U2OS cells 24 h PI with 8 Gy of γ-rays. 1a—Central slices through the nucleus of MCF7 control show SUN1 and nesprin-1 location in the whole nucleus. 2—Convoluted nucleus of U2OS, where one part is formed by one large and one middle-sized MN containing normal chromatin and SUN1, but another part has condensed and fragmented chromatin divided into several MN bordered partially by LA/C and filled by SUN1 (arrows), better seen in 2a—slice showing interrupted LA/C. The arrows in the total image show fragments of SUN1 bordered by LA/C. Two of these fragments also contain heterochromatin (2 arrows). Fragments of SUN1 were probably released from the disrupted nucleus containing heterochromatic residues. Heterochromatin is highly condensed resembling SAHF (arrows at the slice through the image). 3—Convoluted nucleus of MCF7 containing 2 large MN, several blisters, and small MN bordered by LAC and filled by SUN1. Small clumps of SUN1 in the cytoplasm (arrows). 4—MCF7 containing amplified SUN1 in the nucleus, small clumps of SUN1 in the cytosol (arrows), and a cloud of minor dots of SUN1 (thick arrow) accompanied by a dense layer of nesprin-1 dots (long arrow). A clump of nesprin-1 forms a polar cap on the nucleus (arrow). 5—Tweens of MCF7 daughter nuclei connected by a thin anaphase bridge have fragmented SUN1 in the NE (arrows). Fragments of SUN1 are loosened from the NE into the cytosol. Nesprin-1 forms a layer of dots in the NE and surrounds the nuclei. 6—U2OS nucleus with amplified SUN1 and a clump (arrow) containing SUN1 and nesprin-1. Size bar indicates 5µm. (**C**) Examples of SUN2 and nesprin-1, as well as SUN2 and LA/C, mislocalization in MCF7 and U2OS cells at 24 h and 72 h PI with the dose of 8 Gy of γ-rays. 1—MCF7 control nucleus with visualized SUN2 and nesprin-1. 2—MCF7 convoluted nucleus composed of 2 large nuclei in close proximity and 2 MN containing heterochromatin (small arrow in the image of chromatin) and 2 MN with euchromatin (long arrow). In addition, two large fragments of SUN2, containing small fragments of nesprin-1 and heterochromatin (thin arrows) are visible. One of these fragments occurs in the cytosol and the other forms an apical cap. The other of the 2 large nuclei also gained an apical cap formed by a large fragment of nesprin-1 with some small dots of SUN2. 3—Two nuclei connected by a fine anaphase bridge have several MN containing heterochromatin (arrow) or euchromatin bordered by SUN2. Thick arrow shows a clump of SUN2 as apical cap. Nesprin-1 formed tiny dots in the cytoplasm, the NE, and the nuclei. 4—Anaphase bridge binding two MCF7 daughter nuclei. Both nuclei have many MN mostly containing euchromatin and are bordered by SUN2. At the left nuclei is a group of MN filled with SUN2 and nesprin-1 (arrow). A fragment of SUN2 with nesprin-1 also forms a polar cap on the first nucleus and fragments of nesprin-1 are at the surface of the second nucleus. SUN2 is present at an increased level in these nuclei, especially at their narrowing sites at the anaphase bridge. Note, splitting of SUN2 into tiny dots forming clouds around nuclei and MN (arrows in the image showing only SUN2). 5—Convoluted nucleus of U2OS with amplified SUN2 at a part of the nucleus and in MN also containing heterochromatin and nesprin-1. Around the amplified SUN2, a cloud of a tiny dots of SUN2 is formed that is accompanied by a cloud of nesprin-1 dots. 6—Convoluted nucleus of U2OS 72 h PI containing a reinforced layer of SUN2 together with LA/C in several regions of the NE (arrows). In addition to amplified SUN2 in the NE, there are several MN attached to the nucleus as polar caps that are also filled with SUN2. 7—Convoluted nucleus of MCF7 72 h PI composed of 3 large nuclei with several heterochromatic MN attached as polar caps filled with SUN2 and LAC. The level of SUN 2 is reinforced in the NE of the upper nucleus. Size bar indicates µm.

**Figure 6 cells-09-00999-f006:**
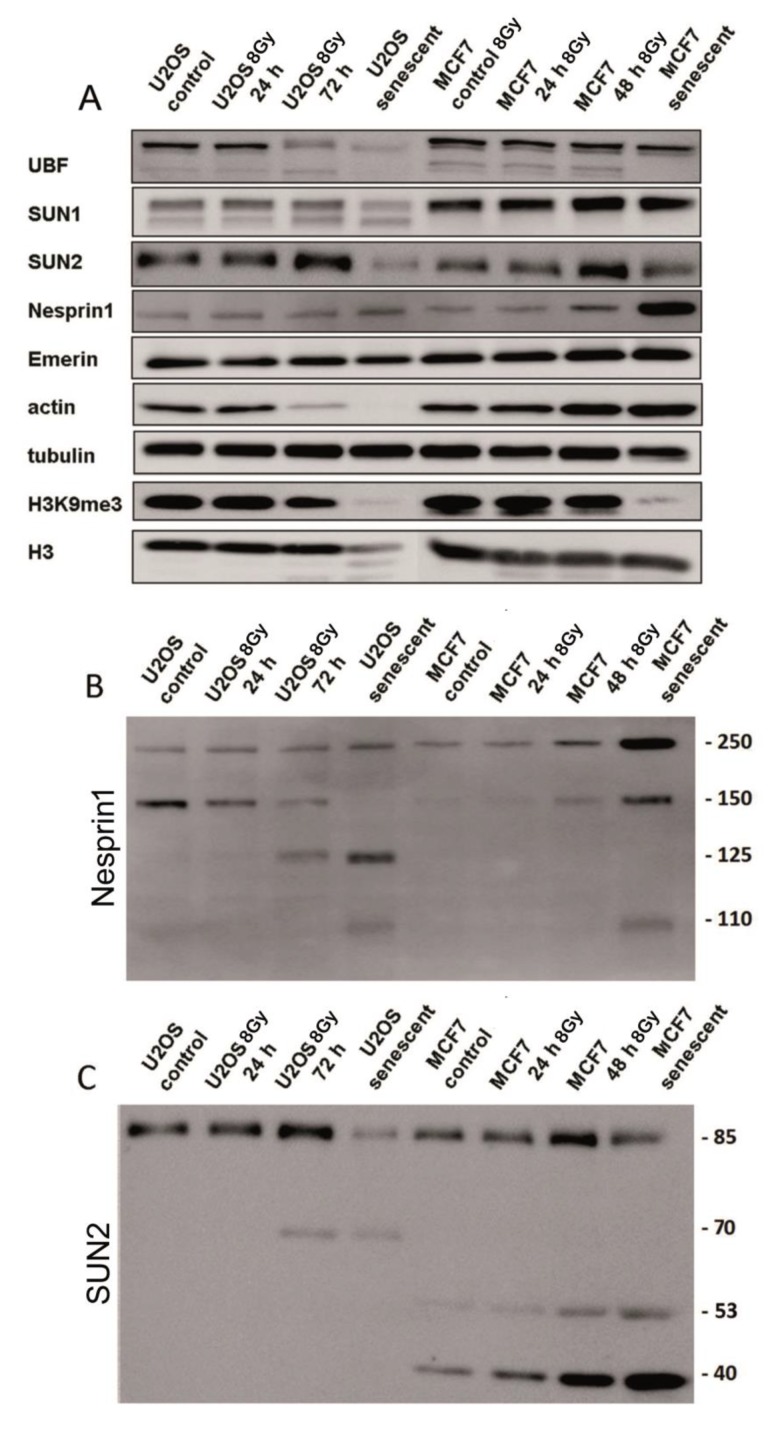
Western blot analysis of protein expression levels (**A**) and their integrity (**B**,**C**) in control and irradiated MCF7 and U2OS cells. (**A**) Blot cuts showing signals of intact proteins. Note, reduction of SUN2 protein and near absence of H3K9me3 epigenetic modification in senescent cells (7 days post irradiation). (**B**,**C**) Reactivity of antibodies against nesprin-1 (**B**) and SUN2 (**C**) proteins on the whole blots. Note, significant fragmentation of intact nesprin-1 (250 kDa band) and SUN2 (85 kDa band) in irradiated cells.

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
