# Peer review of "Spatiotemporal Mislocalization of Nuclear Membrane-Associated Proteins in γ-Irradiation-Induced Senescent Cells"

_cells, 2020, doi:10.3390/cells9040999_

Round 1
Reviewer 1 Report
Using human mammary carcinoma and osteosarcoma cell lines, both wild type and derived shRNA clones deficient in lamin B receptor (LBR) authors investigated redistribution of nuclear membrane associated proteins after inducing senescence by γ-irradiation. The experiments were designed and carried out very carefully, the interpretation of the results are thoughtful, and the results contribute to this research field. Therefore, I recommend this manuscript to be accepted for publication. I have only a couple comments.
1) The title needs modification to better represent the results. What the authors found are not structural changes but re- or mislocalization of the proteins. Therefore, the title should indicate this point. One possible suggestion is: 'Spatiotemporal mislocalization of nunclear membrane associated proteins in γ-irradiation induced senescent cells'.
2) The cell types used are confusing. The term 'clones' make confusion. Authors need to rename the cells such as original cell lines, clones and shRNA knockdown cells. The figure labels need changes accordingly.
Author Response
The experiments were designed and carried out very carefully, the interpretation of the results are thoughtful, and the results contribute to this research field.
Author response: Thank you!
- Comment from Reviewer 1 suggesting : “The title needs modification to better represent the results”
Author response: As suggested by the reviewer, we changed the manuscript title to: “Spatiotemporal mislocalization of nuclear membrane associated proteins in γ-irradiation induced senescent cells”
- Comment of Reviewer 1 suggesting: ”The cell types used are confusing. The term 'clones' make confusion. Authors need to rename the cells such as original cell lines, clones and shRNA knockdown cells. The figure labels need changes accordingly.”
Author response: ”We agree with the reviewer’s suggestion: We renamed the cell lines as “original MCF7 cell line ” and “original U2OS cell line”. Clones derived from these cell lines by treatment with shRNA targeting lamin B receptor (LBR) are renamed as: “MCF7-LBR(-)” and “U2OS-LBR(-) cells”. This terminology is consistently used across the Text, Figures, Tables and Legend to Figures and Tables.
Reviewer 2 Report
The paper I’ve been asked to review describe a very interesting issue related to the modulation of cancer cell proliferation upon gamma irradiation, by a “nuclear membrane proteins” point of view, including morphological and molecular changes.
The authors have followed the dynamics of some of these proteins, including lamin A/C, emerin, SUN1 and 2 and Nesprin 1 and 2.
I think it is a very interesting article that this Journal should accept to publish, however, I have some minor points I would like to deepen with the Authors.
- In the Abstract, the Authors state: “We conclude that structural changes in the LINC complex together with changes in nuclear morphology, uncoupling of heterochromatin from the nuclear membrane and the reduction of LBR and LB1 are early markers of cell senescence”. I think that to conclude this, the authors should first proof that spontaneous senescence is induced by the alteration of expression or functions of one or more of these proteins. So, I think it is more convenient to adjust this statement.
- Line 46. Please specify what Syne 1 and 2 are.
- Figure 2 has never been cited.
- Just out of curiosity, After how long from irradiation, the same cells start to die?
- Line 214. The Authors, refer to DNA damage, obviously induced by gamma irradiation, however, they never show the entity of this, by for example DNA-damage specific markers. It might be interesting to show DNA damage increase over time, together with the appearance of morphological nuclear alterations.
- Line 238/252. The authors refer to “giant cells”. However, in all the figures and tables they talk about “giant nuclei”. Please, they should try to explain to what they are talking about.
- I have a personal doubt. To my experience, U2OS cell line is an extremely heterogeneous cell population, presenting small cells and a portion of giant cell (representing a smaller amount). Didn’t you ever have the perception that the increase of giant cells you see at the final part of the treatment might be due to an earlier death of smaller cells? Did you find any differences in the nuclear lamina proteins localization between these type of cells?
- Figure 6. The Authors should state why they evaluated the expression of UBF and 3mH3K9. What is more, to be clearer, level of expression of Lamin A/C, LBR and LB1 might also be shown.
Minor points regard the use of KESH instead of KASH, LINK instead of LINC, and diverse acronyms given to same proteins, please have a check.
Author Response
Reviewer 2
I think it is a very interesting article that this Journal should accept to publish, however, I have some minor points I would like to deepen with the Authors.
Author response: Thank you for this suggestion. We are pleased to discuss with you some points that are not quite clear in the manuscript in order to make them more understandable.
- Comment of Reviewer 2 suggesting to change the sentence: “We conclude that structural changes in the LINC complex together with changes in nuclear morphology, uncoupling of heterochromatin from the nuclear membrane and the reduction of LBR and LB1 are early markers of cell senescence.”
Author response: Thank you for your suggestion. WE agree that the sentence was not well understood. It now reads as follows: “We conclude that mislocalization of LINC complex proteins is a significant characteristics of cellular senescence phenotypes and may influence complex events at the nuclear membrane including trafficking and heterochromatin attachment”
Comment from Reviewer 2: Line 46. Please specify what Syne 1 and 2 are.
Author response: Syne 1 and Syne 2 are genes coding for spectrin repeat containing proteins Nesprin 1 and Nesprin 2 (large proteins) localizing at outer nuclear membrane. SYNE1 gene is located on chromosome 6q24, SYNE2 on chromosome 14q23. Alternate initiation and splicing of the SYNE genes results in a wide variety of nesprin isoforms. The smaller of which contain variable numbers of spectrin repeats and are truncated at the amino terminus. The smallest isoforms are anchored in the inner nuclear membrane (Zhang et al. 2001). We made this clear in the Introduction.
- Comment from Reviewer 2: Figure 2 has never been cited.
Author response: Thanks you for this notice. This was our mistake. It escaped our attention that when inserting images into the text, five lines of the text in which is the citation to Fig. 2 was lost. We corrected it. The text reads as follows:” The levels of LBR and LB1 decreased significantly already 24 h after the irradiation (24 h PI) (Fig. 1C, D, Table S1A, B). The lower decrease of LBR in U2OS compared to MCF7 cells can be explained by relatively high level of LBR in U2OS cells [23]). At 24 h PI, cells did not express SA-β-galactosidase while they abundantly expressed it at day 7 to day 12 PI (Fig. 2).” Inserted at P.10, line 7 from the top of the chapter No. 3.1
- Comment from Reviewer 2: Just out of curiosity, After how long from irradiation, the same cells start to die?
Author response: Cells were grown attached to the bottom of the vessel, immersed in growth medium, exchanged every second day maximally 3 weeks after irradiation. During this time they remained attached to the bottom; only rare cells peeled off and were eliminated when changing medium. These cells died by apoptosis. Cells attached to the bottom gradually increased the size of the cytoplasm forming many protrusions by means of which they associated with neighboring cells and formed very nice patterns. Approximately, after 3 weeks we stopped exchange medium, the cells began to detach from the bottom and died. May be, that if we exchanged the medium for longer period, these senescent cells would stayed alive longer.
- Comment from Reviewer 2: Line 214. The Authors, refer to DNA damage, obviously induced by gamma irradiation, however, they never show the entity of this, by for example DNA-damage specific markers. It might be interesting to show DNA damage increase over time, together with the appearance of morphological nuclear alterations.
Author response: In the past, we studied the repair of DSB induced by γ-irradiation with different doses and under different conditions several times. We found that many DSB were repaired during 24 h PI, but some of them remained long time PI and their presence was important for transfer of cells to senescence. During senescence new breaks appeared and it was not easy to distinguish them from original ones. The number of DSB remaining non-repaired 24 h PI depended on the radiation dose used on the cell line. With this in mind, we did not consider it important to detect non-repaired DSB during monitoring changes in nuclear membrane proteins. But it could be a nice theme for diploma thesis.
- Comment from Reviewer 2: Line 238/252. The authors refer to “giant cells”. However, in all the figures and tables they talk about “giant nuclei”. Please, they should try to explain to what they are talking about.
Author response : Almost all cells had a large cytoplasm with many large protrusion already 24h PI (typical giant senescent cells), however this cytoplasm could be visible only when we detected emerin, because this protein was dispersed in the cytoplasm, filled it completely together with all its protrusions so that the entire cell size could be seen. However, we mainly detected LINC complex proteins that were not dispersed to the cytoplasm as emerin and therefore we were not able to observe whole cells – nuclei together with cytoplasm by our confocal microscope in all followed cells. Therefore we focused on monitoring giant nuclei that occurred in U2OS but not in MCF7 cell line. These giant nuclei were more than 20 times larger compared to nuclei of this control cell line and could be easily distinguished. The giant nuclei were bounded by lamin A/C, similarly as nuclei of other cells. Lamin A/C was degraded on rare giant nuclei. Examples are shown at Fig. 5C and S2C.
At the p. 11, line 6 from the bottom we explained why we followed giant nuclei Instead of giant cells.
“We preferred to follow giant nuclei, because it was difficult to track giant cells in our conditions.”
Comment from Reviewer 2: I have a personal doubt. , U2OS cell line is an extremely heterogeneous cell population, presenting small cells and a portion of giant cell (representing a smaller amount). Didn’t you ever have the perception that the increase of giant cells you see at the final part of the treatment might be due to an earlier death of smaller cells? Did you find any differences in the nuclear lamina proteins localization between these type of cells?
Author response: We thank the reviewer for the interesting idea. Certainly, the U2OS cells are not phenotypically homogeneous and their morphologies differ depending on culture confluency and cell cycle. We a priori cannot exclude the possibility that giant cells are selected after the irradiation following death of small cells. However, the number of giant cells was relatively high (10-20 x fold compared to actively proliferating cells) at 24 hours post irradiation. At this interval we have no evidence for cell death since nearly all cells tend to adhere to dish surface. Extracted DNA does not show signs of degradation suggesting that apoptosis has not been activated yet. We therefore incline to interpretation that giant nuclei arose as a consequence of DNA damage and could be a prerequisite towards formation of micronuclei and other cellular aberrations. It is also possible that polyploidy Is going on in irradiated cells and that the giant cells contain endoreduplicated DNA.
- Comment from Reviewer 2: . The Authors should state why they evaluated the expression of UBF and 3mH3K9. What is more, to be clearer, level of expression of Lamin A/C, LBR and LB1 might also be shown.
Author response:
In addition to LINC proteins the Western blot in Figure 6 contains a set of proteins which were previously shown to be involved in cellular responses to irradiation. For example, UVA irradiation decreased the levels of H3K9me3 and H4K20me3 at 28S rDNA (Stixova et al. Chromosome Research 27, 41–55, 2019). Here we are showing that the H3K9me3 levels are decreased almost to negligible levels in senescent cells. Since H3K9me3 is a heterochromatic epigenetic mark its loss could be explained by structural changes caused by detachment and distension of heterochromatin from NM in senescent cells (Lukasova et al. 2017). We discussed this possibility on page 18, first para, towards the end.
We agree with the reviewer about the comment on the lack of Lamin A/C, LBR and LB1 immunoblots. The LA/C, LBR and LB1 antibody reactivity was already shown in our previous work (Figure 3E). However, we agree with the reviewer that it is important to show that the shRNA LBR transgenes are functional even after prolonged cultivation of cells. Therefore, we repeated the blot and immunostained cell extracts from the original lines and their shRNA LBR clones (new Figure S3) with antibodies to LBR, SUN2 and nesprin-1. It is evident that both MCF7-LBR(-) and U2OS-LBR(-) exhibited faint LBR and LB signals on blots compared to the original lines.